# Analyzing the Urban-Rural Vaccination Coverage Disparity through a Fair Decomposition in Zhejiang Province, China

**DOI:** 10.3390/ijerph16224575

**Published:** 2019-11-19

**Authors:** Yu Hu, Ying Wang, Yaping Chen, Hui Liang

**Affiliations:** Institute of Immunization and Prevention, Zhejiang Provincial Center for Disease Control and Prevention, Hangzhou 310051, China; ywang@cdc.zj.cn (Y.W.); ypchen@cdc.zj.cn (Y.C.); hliang@cdc.zj.cn (H.L.)

**Keywords:** vaccination, coverage, inequality, decomposition, urban, rural

## Abstract

*Objectives*: Exploring determinants underlying disparities in full vaccination coverage (FVC) can contribute to improved immunization interventions. FVC and its risk factors in Zhejiang province have been studied, yet the determinants explaining the rural–urban disparity in FVC have not been studied. This study aimed to disentangle the factors explaining rural–urban disparities in FVC of vaccine doses scheduled during the first year of life in Zhejiang province. *Methods*: We used data from a vaccination coverage survey among children aged 24–35 months conducted in 2016. The outcome measure was full vaccination status, and the grouping variable was the area of residence. Descriptive statistics were used to analyze the FVC and rural–urban residence across the exposure variables. The Fairlie decomposition technique was used to decompose factors contributing to explaining the FVC disparity. *Results*: There were 847 children included in this study, of which 49.6% lived in a rural area. FVC was 94% in rural areas and 85% in urban areas. A disparity of 9% to the advantage of the rural areas and the exposure variables explained 81.1% of the disparity. Maternal factors explained 49.7% of the explained disparity with education, occupation, and ethnicity being the significant contributors to the explained disparity. Children’s birth order and immigration status contributed somewhat to the explained inequality. *Conclusion*: There was a significant disparity in FVC in Zhejiang province, a disadvantage to the urban areas. Policy recommendations or health interventions to reduce the inequality should be focused on eliminating poverty and women’s illiteracy, targeted at migrant children or children from minority ethnicities.

## 1. Introduction

A remarkable success in vaccination coverage has been observed globally, but significant disparities still exist between or within countries [1,2,3]. High vaccination coverage needs to be attained and maintained for all required antigens, known as full vaccination coverage (FVC) in all communities to ensure a sustained and continued decrease in the incidence of vaccine preventable diseases (VPDs). The analysis of determinants explaining the difference in vaccination coverage between communities can help increase the FVC by decreasing the inequalities in coverage. Determinants influencing vaccination coverage have been widely studied in developing countries, and similar risk factors were found to explain both vaccination coverage and disparities therein. Generally, socioeconomic and demographic characteristics of mothers or children, accessibility to vaccination service, information, and belief-related factors have been observed as determinants of the vaccination coverage [4,5].

Zhejiang province, located in east China, is one of the most developed provinces in the mainland of China, with a population of 72 million based on the 2015 census. Zhejiang province consists of 11 cities, 90 counties, and 1319 towns, with a geographic area of 101,800 km^2^. About 62.3% of the total population lives in urban areas and the rest in rural areas. The Zhejiang provincial expanded program on immunization (EPI) was launched in 1978, and it had successfully reduced childhood morbidity and mortality. The EPI’s success is believed to be driven by feasible access to vaccination service and the parents being receptive to vaccination. Although Zhejiang province has an effective and efficient immunization program, some perplexing variations within the province still exist. Zhejiang province has attained and maintained high coverage (>90% for individual vaccine doses) for almost all the individual antigens in the national vaccination schedule as per administrative reports [6,7], but two rounds of provincially representative surveys (2011 and 2013) have found the surprising gaps in coverage between rural and urban areas to the advantage of rural areas [8]. In Zhejiang province, vaccination services are delivered through fixed clinic sessions conducted at community health centers and all vaccines included in EPI are provided free of charge. The primary vaccination schedule before the first birthday includes: Bacillus Calmette-Guerin (BCG) at birth; hepatitis B vaccine (HBV) at birth, 1 month, and 6 months; oral poliomyelitis vaccine (OPV) at 2, 3, and 4 months; diphtheria, tetanus, acellular pertussis combined vaccine (DTaP) at 3, 4, and 5 months; measles and rubella combined vaccine (MR) at 8 months [9]. 

In China, Min Lv [10] had reported that the vaccination coverage was higher among people living in rural areas compared with urban areas, even under the free vaccination policy. This report also indicated that family income, health education and promotion, and gender were the associated factors of the observed disparity. Vaccination coverage, timeliness, cost-effectiveness, and the impact of selected vaccines in Zhejiang province have been studied, as well as the risk factors associated with timeliness and coverage [11]. However, the knowledge regarding determinants that explained the disparity in vaccination coverage between rural and urban areas in Zhejiang province is still lacking. Such an analysis could be of interest worldwide, shedding some light on complexities of vaccination service delivery and the basic health system, which may improve vaccination coverage, close the coverage gaps between urban and rural groups, and finally drive equalities in vaccination service [12]. Thus, this study aimed to disentangle the factors explaining rural–urban disparities in FVC in Zhejiang province by second use of the 2016 provincial coverage survey among children aged 24–35 months, for the vaccine doses scheduled during the first year of life. 

## 2. Methods

### 2.1. Data Source and Study Subjects

The data source of this study was the provincially representative vaccination coverage survey implemented in 2016. Between 15 and 20 November 2016, a household-based cluster survey among children aged 24–35 months (born from 1 December 2013 to 1 November 2014) living in Zhejiang province was conducted. This age range was chosen thus that all of the eligible children had the chance of having a complete primary vaccination scheduled before their first year of life. 

### 2.2. Sample Size

The sampling procedure was based on the immunization cluster survey recommended by WHO [13]. The formula used to estimate the sample size was as follows:(1)$$N_{\min}=deff\times \frac{z_{\left(1-\alpha /2\right)}^{2}\times p\times \left(1-p\right)}{d^{2}}$$

To reach the estimates of coverage at city level with a two-tailed α error of 5% and a permissible error (*d*) of 0.08, assuming the expected timely vaccination coverage (*p*) at 85% and a design effect (*deff*) of 2, the minimum sample size required for each city was 77 eligible children, divided into 11 clusters (towns) of 7, corresponding to 847 children in the entire province.

### 2.3. Field Investigation

The procedure of field investigation included 3 steps: First, 11 towns were selected in every city by using the probability proportional to population size method and the population size of each town from the census 2015. Second, the index community was chosen by drawing lots from the list of community of each selected town. Similarly, the index household was chosen from the selected community by using the random number table. Third, the adjacent household on the right of the index household was investigated when the survey of the previous household was done, and the rest of the households were investigated in the same way. If there was more than one eligible child in the household, the one whose birthday was close to the investigation day was chosen. Households where somebody lived but without any response were re-visited until they could not be contacted for another 2 attempts. If there were not 7 eligible children in the selected community, the closest community was chosen to investigate the remaining samples. 

### 2.4. Data Collection

Mothers of the eligible children were surveyed by EPI staff from the 11 centers for disease control and prevention (CDCs) at city level, and all of them attended a one-day training before the fieldwork. The training course included the purposes of the survey, the content of the questionnaire, the procedure of the investigation, and the skill of investigating sensitive items. All investigators were not responsible for the investigation sites where they normally worked to avoid selection bias.

A standardized, pretested questionnaire was used for face-to-face interviews with the mother of each eligible child. Information on the children, their mothers, and the sociodemographic of their households was collected. Vaccination history was transcribed from the immunization cards and was validated through Zhejiang provincial immunization information system. The responses were categorized into those who said it was a big problem and those who said it was not a big problem.

### 2.5. Measurements

Full vaccination status was derived from the child’s vaccination history as a dichotomous variable. The target child was considered as fully vaccinated if he/she had received the 11 vaccine doses scheduled before 1 year of age as we mentioned before. The status of residence (urban or rural) was the variable used to categorize the surveyed children. The exposure variables were selected based on the literature and induced into 3 aspects: Child’s, mother’s, and sociodemographic. The children’s variables included child’s gender and the child’s birth order; the maternal variables included the mothers’ ages, mother’s education, mother’s occupation, mother’s ethnicity; the sociodemographic variables included monthly household income per capita, immigration status, and the distance to the health facility. The categories of these variables were entered as dummy variables in the analyses, as shown in Table 1.

### 2.6. Data Analyses

The Fairlie decomposition technique [14], which was an extension of the Blinder-Oaxaca for logistic and probit models implemented in Stata software using the module developed by Jann, was used to decompose the disparity in FVC of children between rural and urban areas. The Fairlie decomposition technique tested how much of the difference in FVC of children between rural and urban areas could be explained by rural–urban differences in variables included in this study. Furthermore, it can also estimate the contribution of each variable to the explained FVC difference between rural and urban areas. A positive coefficient would result in a positive contribution to the inequality, and it was interpreted as supporting (increasing) the inequality. Conversely, a negative coefficient would yield a negative contribution to inequality and consequently decreases inequality.

Descriptive analyses of FVC and exposure variables were implemented. The logistic regression model was used to estimate the individual variable’s effect on FVC as well as report the marginal effects, their standard errors, and *p* values. After that, the two decomposition models were fitted. First, we estimated the rural–urban FVC disparity and the explained difference, and explained the individual contribution of each exposure variable to the explained difference. Second, the rural–urban FVC disparity and the explained difference was estimated, and the contribution of each 3 groups of exposure variables (child’s, mother’s, and sociodemographic) to the explained difference was measured. 

All analyses were performed using STATA 14.0 (StataCorp. 2015, Stata statistical software, college station, TX, USA) at a two-tailed significance level of α of 0.05. We used survey weights through the STATA ‘‘svy” command in our analyses. 

### 2.7. Ethical Considerations

This study was approved by the ethical review board of ZJCDC (T-047-D). Written informed consent was obtained from each mother once there was a decision to participate.

## 3. Results

Table 2 showed the proportions of different levels of the exposure variables by residence status. In rural areas, there was a significant overrepresentation of higher birth order numbers and migrant children. For maternal variables, in rural areas, there was an over-representation of women who were under 30 years of age, with lower education levels, with no job, from minority ethnics, and from households with a lower monthly income. 

There was 847 weighted number of eligible children included in this study, of which 49.6% lived in a rural area. The FVC across the exposure variables was presented in Table 1. FVC was higher in rural areas, among children with higher birth order, children born to mothers aged above 30 years, children of mothers with senior middle school education level or above, children of mothers with a job, among resident children, and among children from households with the higher monthly income per capita. In terms of ethnic, the Han groups had a higher proportion of FVC. 

In the multivariable regression model, the determinants that significantly and positively associated with FVC included birth order, mother’s age, mother’s education level, and mother’s having a job compared to being home fulltime. Besides, migrant children and minority ethnicity were significantly and negatively related to FVC (Table 3). 

The rural–urban disparity in FVC was 9.0%. Of them, 7.3%, representing 81.1%, was explained by differences in the exposure variables included in this analysis. Of them, maternal age contributed 49.7%, maternal education contributed 24.0%, maternal occupation contributed 32.7%, maternal ethnicity contributed −36.3%. Children’s birth order explained 26.3% of the rural–urban disparity in FVC, while children’s gender had no significant contribution to the FVC disparity. Among the sociodemographic factors, immigration status contributed −54.4%, while distance to health facility and household income per capita did not contribute significantly to explaining the rural–urban immunization disparity (Table 4).

## 4. Discussion 

This study indicated that there was FVC inequality between urban and rural areas, in that higher FVC was observed among children in rural areas, as shown in previous reports worldwide [15,16,17]. Achieving high FVC in rural areas demonstrated that the vaccination service was accessible even to rural communities in Zhejiang province. Most of the disparity could be explained by factors from the child’s mother but in somewhat different directions. Child’s birth order, household income, and immigration status were also contributors to explaining the inequality. However, the perceived distance to a health facility was not an explaining factor.

More optimal FVC observed in rural areas compared to urban areas was a notable trend since many pieces of research have reported rural residents having significantly lower utilization of public health services across many developing countries [18,19]. We stipulated that the difference in service delivery mechanism between rural and urban areas might be the main reason for the disadvantage of vaccination service in urban areas. As common sense, caregivers consider the vaccination worker as an authoritative source of information on vaccination, and their explanation is a good opportunity to correct misunderstanding or rumors on vaccination, which can encourage caregivers to get the vaccination service completed [10]. Specifically, vaccination workers in rural areas could more easily spread knowledge on immunization and encourage parents to bring their children to get vaccinations through their closer relationship with communities. On the other hand, the rapid pace of urbanization occurred due to high-speed economic development in last three decades in the Zhejiang province, which attracted more rural population flow into urban areas and put a heavy workload on health workers. Most vaccination workers working in urban immunization clinics did not have time to explain the importance of vaccination to parents. The lack of direct connection between parents and providers made it difficult to provide personal contact and health education to residents. Crowd problems and long waiting times at immunization clinics would also be obstacles to vaccination in urban areas.

Of the factors in regards to the child, birth order was found to be a risk factor contributing to the inequity in FVC, with children of higher birth order having the higher FVC. However, our findings differed from studies carried out, for example, in the Philippines and the U.S., where children were less likely to be fully vaccinated if the birth order was higher [20,21]. These two previous reports stated that the allocation of the family resources, which specifically meant the parental initiative and the dedication of time in addition to the financial expense, would be preferred to the younger child in the household. This association could not be verified as the relevant attitude of the mothers was not investigated. We assumed that mothers who had more children should have more experience in raising children and would be more aware of childhood immunization and be more familiar with the vaccination schedule. The experience of raising a child would help mothers to vaccinate their children completely. 

Maternal factors were significant determinants to explain the inequity of the urban–rural difference in FVC. This study found that there was a significant association between the mother’s age and FVC, with children with an older mother having a higher FVC. The possible explanation was that the older mother would have better utilization of public health care, leading to better accessibility and utilization of vaccination services [22]. Another explanation was that older mothers would have the possibility of having more than one child, which might lead to having better experience in raising children. Maternal education is a common determinant of vaccination status globally, with higher coverage usually found with higher education levels [23,24,25,26]. The results from the decomposition analysis were consistent with the previous reports. It was demonstrated that higher maternal education levels could make mothers have a better understanding of vaccination schedules and practice through enhancing the accessibility of information on vaccination, as well as facilitating the communication between mothers and vaccination workers. Hence, we recommended that it will be important to improve the awareness of vaccination among mothers, through establishing a health education mechanism towards mothers on the necessity of full vaccination, except for using the traditional education system. Most surprisingly, it had been indicated in previous studies that children with non-working mothers might have a higher possibility of full vaccination [27,28,29,30,31,32]. However, we found children born to employed mothers had a higher FVC. The possible reason was that working mothers might have more financial resources and a stronger position within the household. That meant they would be more empowered, have a higher awareness, and have solid finances to support childhood vaccination. Our findings suggested that empowering women, for example by encouraging them to take up jobs, should have the potential to not only improve their livelihood but also possibly increase full vaccination of their children. Minority ethnicity was found as a negative factor of FVC. It was in line with a previous report [33], which stated ethnicity was a risk factor of health-seeking behaviors and outcomes. Mothers with minority ethnicities were considered as marginalized populations in China. Most of them had lower education and socioeconomic status compared to Han ethnicity, leading to poor awareness of primary health care, including childhood vaccination. Thus, it was necessary to pay special attention to these disadvantages in sub-populations to increase accessibility and improve the awareness of vaccination. 

The relationship between the migrant people and the inequity of the urban–rural difference in FVC has been well established in Zhejiang province [11]. We assumed that the migrants might face challenges of adapting to the new socio-cultural environment. On the contrary, the residents could avail themselves of the vaccination service much better since they are familiar with the living areas. Distance to vaccination sites is a commonly cited reason for under-vaccination [34]. However, the FVC was not significantly different between mothers who perceived it was a big problem, and those did not. Furthermore, it also did not contribute to explaining the differences in FVC. This might be due to the high density of immunization clinics, which improved the accessibility of vaccination services in both rural and urban areas. Moreover, the gap in traffic convenience between rural and urban areas had been narrowed due to the high-speed road construction in rural areas. Another reason for our finding was that the measurement was for the subjective perception of distance from the home to the health center. Hence, mothers would consider an actual long-distance as short if the traffic was very convenient. 

Our results had interesting implications for vaccination programs both in Zhejiang province and internationally. Sufficiently, complete vaccination coverage not only protects vaccinated individuals but also can protect even the unvaccinated, and even in the long run, would eliminate or eradicate many infectious diseases. Hence, it is necessary to know the ever-evolving factors that drive or limit the utilization of vaccination services in different settings or sub-populations. Our results stressed that more research was needed to figure out what risk factors were driving lower FVC in urban areas. Meanwhile, we also suggested that it was time to reconsider interventions in urban areas to close the FVC gap between urban and rural areas.

There were several limitations in this study. First, since we used the secondary data of the 2016 Zhejiang provincial coverage survey, the reasons for under-vaccination were not investigated but deserve assessment in future researches. We believed that these variables would influence our results, for example, by increasing the explained difference. Second, data limitation might be very likely as some children could not be vaccinated due to underlying diseases such as being immunocompromised. It was not accounted for in both surveys and subsequent analysis, however, we considered that it should not affect the results significantly as such underlying diseases were very rare.

## 5. Conclusions

A substantial disparity in FVC was observed between rural and urban areas, to the disadvantage of the urban areas. The disparity is mostly explained by the child’s and mother’s, as well as the socioeconomic factors. Policy recommendations or health interventions to reduce the inequality in FVC between urban and rural areas should be focused on eliminating poverty and women’s illiteracy, targeted at migrant children or children from minority ethnicities. Those interventions will not only close the gaps of FVC between urban and rural areas but also improve the sustainable development of EPI in Zhejiang province.

## Figures and Tables

**Table 1 ijerph-16-04575-t001:** Weighted full vaccination coverage across all the variables (*n* = 847).

Variables	Level	Full Vaccination Coverage		*p* *
		Yes *n* (row, %)	No *n* (row, %)	
Residence	Rural	395 (94.0)	25 (6.0)	<0.001
	Urban	363 (85.0)	64 (15.0)	
Children’s aspect				
Child’s gender	Male	380 (89.6)	44 (10.4)	0.901
	Female	378 (89.4)	45 (10.6)	
Birth order	1	435 (87.0)	65 (13.0)	0.005
	2	237 (91.5)	22 (8.5)	
	≥3	86 (97.7)	2 (2.3)	
Material aspect				
Mothers’ ages	<30	508 (86.1)	82 (13.9)	<0.001
	≥30	250 (97.3)	7 (2.7)	
Mother’s education	< senior middle school	126 (80.3)	31 (19.7)	<0.001
	≥ senior middle school	632 (91.6)	58 (8.4)	
Mother’s occupation	Home fulltime	222 (82.2)	48 (17.8)	<0.001
	Employed	536 (92.9)	41 (7.1)	
Mother’s ethnicity	Han	718 (92.2)	61 (7.8)	<0.001
	Minority	40 (58.8)	28 (41.2)	
Sociodemographic aspect				
Household income per capital	<5000 CNY/month	228 (89.4)	27 (10.6)	0.245
	5000–10000 CNY/month	327 (87.9)	45 (12.1)	
	>10000 CNY/month	203 (92.3)	17 (7.7)	
Immigration status	Resident	486 (95.5)	23 (4.5)	<0.001
	Migrant	272 (80.5)	66 (19.5)	
Distance to health facility	Not a big problem	539 (89.7)	62 (10.3)	0.776
	A big problem	219 (89.0)	27 (11.0)	

*: *χ*^2^ test was used with a two-tailed significance level of *α* of 0.05.

**Table 2 ijerph-16-04575-t002:** Weighted frequencies of exposure variables by residence (*n* = 847).

Variables	Level	Residence		*p* *
		Rural *n* (col, %)	Urban *n* (col, %)	
Child’s gender	Male	212 (50.5)	212 (49.6)	0.810
	Female	208 (49.5)	215 (50.4)	
Birth order	1	218 (51.9)	282 (66.0)	<0.001
	2	140 (33.3)	119 (27.9)	
	≥3	62 (14.8)	26 (6.1)	
Mothers’ ages	<30	317 (75.5)	273 (63.9)	<0.001
	≥30	103 (24.5)	154 (36.1)	
Mother’s education	< senior middle school	118 (28.1)	39 (9.1)	<0.001
	≥ senior middle school	302 (71.9)	388 (90.9)	
Mother’s occupation	Home fulltime	179 (42.6)	91 (21.3)	<0.001
	Employed	241 (57.4)	336 (78.7)	
Mother’s ethnicity	Han	372 (88.6)	407 (95.3)	<0.001
	Minority	48 (11.4)	20 (4.7)	
Household income per capital	<5000 CNY/month	159 (37.9)	96 (22.5)	<0.001
	5000–10000 CNY/month	167 (39.8)	205 (48.0)	
	>10000 CNY/month	94 (22.3)	126 (29.5)	
Immigration status	Resident	202 (48.1)	307 (71.9)	<0.001
	Migrant	218 (51.9)	120 (28.1)	
Distance to health facility	Not a big problem	301 (71.7)	300 (70.3)	0.652
	A big problem	119 (28.3)	127 (29.7)	

*: *χ*^2^ test was used with a two-tailed significance level of *α* of 0.05.

**Table 3 ijerph-16-04575-t003:** Weighted marginal effects of multivariable logistic regression of full vaccination on the exposure variables (*n* = 847).

Variables	Level	dy/dx ^1^ (SE ^2^)	*p*
Child’s gender	Male	Ref	
	Female	0.051 (0.062)	0.452
Birth order	1	Ref	
	2	0.025 (0.023)	0.039
	≥3	0.093 (0.035)	<0.001
Mothers’ ages	<30	Ref	
	≥30	0.073 (0.005)	0.012
Mother’s education	< senior middle school	Ref	
	≥ senior middle school	0.072 (0.029)	0.019
Mother’s occupation	Home fulltime	Ref	
	Employed	0.086 (0.032)	<0.001
Mother’s ethnicity	Han	Ref	
	Minority	−0.092 (0.016)	0.002
Household income per capita	<5000 CNY/month	Ref	
	5000–10000 CNY/month	0.037 (0.101)	0.013
	>10000 CNY/month	0.048 (0.065)	<0.001
Immigration status	Resident	Ref	
	Migrant	−0.113 (0.003)	<0.001
Distance to health facility	Not a big problem	Ref	
	A big problem	<0.001 (0.431)	0.925

Note: ^1^: Marginal effect; ^2^: Standard Error.

**Table 4 ijerph-16-04575-t004:** Weighted decomposition of the disparity in full vaccination coverage between rural and urban residences (*n* = 847).

Variable	Value/Coefficient ^1^	% Contribution to the Explained Disparity	*p*
Rural full immunization coverage	94.0%		
Urban full immunization	85.0%		
Rural–Urban disparity	9.0%		
Total explained disparity	7.3%		
% total explained disparity	81.1%		
Children’s aspect			
Child’s gender	0	0.0	0.992
Birth order	0.0045	26.3	0.017
Subtotal	0.0045	26.3	0.017
Material aspect			
Mothers’ ages	0.0085	49.7	<0.001
Mother’s education	0.0041	24.0	0.026
Mother’s occupation	0.0056	32.7	0.004
Mother’s ethnicity	−0.0062	−36.3	0.002
Subtotal	0.011	64.3	<0.001
Sociodemographic aspect			
Household income per capital	0.0006	4.6	0.233
Immigration status	−0.0093	−54.4	<0.001
Distance to health facility	0.0017	9.9	0.080
Subtotal	0.0009	5.3	0.152

^1^: Two models were used to decompose the results presented above. The first model decomposed the rural–urban disparity using individual variables, and the second model decomposed the disparity using theoretical perspective grouped (children’s, material, and sociodemographic) dummy variables.

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
