# Peer review of "Analyzing the Urban-Rural Vaccination Coverage Disparity through a Fair Decomposition in Zhejiang Province, China"

_ijerph, 2019, doi:10.3390/ijerph16224575_

Round 1
Reviewer 1 Report
This is a well conceived and well executed study. For an English reader however, there are a number of changes that should be made to the manuscript to clarify and provide ease of reading. For example line 38 should provide a pleural of the word "province". Line 59 should have the word "lack" changed to "lacking". There are number of places where the indefinite articles a/an should be placed, e.g. "a child from a working mother", not just "child from working mother". There are a number of instances in the discussion where the English translation is close but not quite hitting the mark. For example in lines 187–189, "completed" should be used instead of "completely"; "easily spread" should be changed from the statement "not convenient to spread";206 "The experience from raising a child " as opposed to "These accumulated rich experience "; 210 and 211 "an" older mother; "might lead to having a better experience" as opposed to "might lead to get a better experience", 254 "more research "not more researches "; 260 a "difference", not "differential"; 262 it was now accounted for "in" both survey and analysis….
I am sure that there are other translation differences but those are the ones that struck me in reading.
Author Response
Response: These language problems had been fixed according to your comments.
Reviewer 2 Report
The investigation examined the factors explaining rural-urban disparities in full vaccination coverage children in Zhejiang. The manuscript is generally well-written and I just got several minor comments:
The chi-square statistics do not to be reported in tables as p-values have already been used. "Logistic regression", not "logit regression" I would like to know how the authors measure "Distance to health facility" this variable. Why did father's information not collected? Seems many similar studies have been done before. what are added into literautre? If just looking at the conclusion, it is just a very general recommendation.Author Response
Response: We deleted the chi-square statistics in the table 1. We changed the word logit as logisitic. This question was used as a proxy for distance to health facility. The responses were categorized into those who said it was a big problem and those who said it was not a big problem. According to our study protocol, we just surveyed children’s mothers as they would be more familiar childhood care service than fathers.
Reviewer 3 Report
Abstract: Abstract is lack of an introduction, please provide a brief introduction in the abstract section. Also the redaction of results section in the abstract is not clear. Introduction: Line 54: the acronym for diphtheria, pertussis and tetanus is incomplete. In your province, which vaccine did you have DPaT or DPwT? In the introduction it is necessary to express the reason for doing this study. What are the reasons that generated the research question? Also, what factors the authors believe that can cause the reduction of vaccination coverage? Material and metods: Statistical test employed to analyzed data are lack. Results: In all tables, which statistical tests were performed? Urban and rural subjects seems to be very different population, therefore I suggest tu change the order of the table 1 by the table 2 and vice versa. Discussion and conclusions: none. It is clear that the location of the patients (urban or rural) is a confusing factor. Therefore, in the table 3, the multivariable logit regression need to include as co-variable the location of the patients. In table 3, please reportes instead of SD, 95%IC. Also it will be useful, that the authors also calculate the OR of the variables included in table 3.
Author Response
Abstract: Abstract is lack of an introduction, please provide a brief introduction in the abstract section. Also the redaction of results section in the abstract is not clear. Introduction:
Response: we added some background information and main results in abstract.
Line 54: the acronym for diphtheria, pertussis and tetanus is incomplete. In your province, which vaccine did you have DPaT or DPwT?
Response: In China, DTaP is included in the EPI schedule.
In the introduction it is necessary to express the reason for doing this study. What are the reasons that generated the research question? Also, what factors the authors believe that can cause the reduction of vaccination coverage?
Response: we added a literature from Beijing to describe the disparity in vaccination coverage found in China in recent years.
Material and metods: Statistical test employed to analyzed data are lack.
Response: we wrote this section in 2.6 part.
Results: In all tables, which statistical tests were performed? Urban and rural subjects seems to be very different population, therefore I suggest tu change the order of the table 1 by the table 2 and vice versa.
Response: we changed the order of table 1.
Discussion and conclusions: none. It is clear that the location of the patients (urban or rural) is a confusing factor. Therefore, in the table 3, the multivariable logit regression need to include as co-variable the location of the patients. In table 3, please reportes instead of SD, 95%IC. Also it will be useful, that the authors also calculate the OR of the variables included in table 3.
Response: in table 3, we just wanted to calculate the marginal effects of individual variable’s contribution on FVC, which would be used to estimate their relevant contribution to the disparity between urban and rural areas. As such, there would be no necessity to include the location of participants as a co-variable. After that, the decomposition model was fitted to assess the these risk factor’s contribution on the disparity between urban and rural areas.